# More is more: leveraging multi-rater information for whole slide images grading via virtual expert panel

**Jan Grove**[1,2] (iD)      JANGROVE.JG@GMAIL.COM

**Michel Botros**[1,2,3,4] (iD)      M.BOTROS@AMSTERDAMUMC.NL

**Ylva A Weeda** [1,3,4] (iD)      Y.A.WEEDA@AMSTERDAMUMC.NL

**Clara I. Sánchez** [1,2] (iD)      C.I.SANCHEZGUTIERREZ@UVA.NL

**Erik Bekkers**[1] (iD)      E.J.BEKKERS@UVA.NL

**Sybren L Meijer**[3,4] (iD)      S.L.MEIJER@AMSTERDAMUMC.NL

**Hoel Kervadec**[1,2] (iD)      H.T.G.KERVADEC@UVA.NL

[1] *qurAI, Informatics Institute, University of Amsterdam*

[2] *Amsterdam University Medical Center at the University of Amsterdam, department of Biomedical Engineering and Physics, Amsterdam, The Netherlands*

[3] *Amsterdam University Medical Center at the University of Amsterdam, department of Pathology, Amsterdam, The Netherlands*

[4] *Cancer Center Amsterdam, Cancer Treatment and Quality of Life*

**Editors:** Accepted for publication at MIDL 2026

## Abstract

In medical imaging, datasets with several expert diagnoses capture diagnostic uncertainty, yet many approaches compress diagnoses into a single consensus label. Due to its highly subjective nature, Barret's Esophagus gradings often diverge, thus necessitating several expert opinions to mitigate variation in diagnostic or treatment outcomes. Using a multi-rater dataset from the Dutch Esophageal Pathology Panel, we propose an approach to tackle the implied issues such as poor calibration and overconfident predictions that come with a compressed label. We offer an approach that models individual rater behaviors as part of virtual panels, allowing for better prediction performance while also improving the quality of uncertainty estimates for clinical decision-making when compared to pre-compressed labels. We show that due to their individual correlation with the clinical consensus, a combination of raters—especially an inclusion of all raters—yields higher performance and better calibrated predictions. The code is available at https://github.com/jangrove379/WeakBE-Net.

**Keywords:** Multi-rater Learning, Multiple Instance Learning, Uncertainty Estimation, Histopathology, Barrett's Esophagus

## 1. Introduction

Medical imaging studies increasingly rely on large datasets annotated by multiple experts. While this provides a richer view of clinical knowledge, it also exposes a fundamental challenge: expert labels often disagree, sometimes substantially. Differences in training, experience, background information, and subjective interpretation all contribute to variability in diagnostic settings assessing medical images like pathology, radiology, and dermatology. This variability is not noise but a reflection of the underlying diagnostic uncertainty. Yet most deep learning models reduce multi-rater labels into a single consensus or majority vote,

discarding valuable information about the ways in which experts disagree and the extent of those disagreements.

Training deep learning models on these collapsed labels implicitly assumes that a single "correct" label exists for each case, even when the medical community itself does not fully agree. This simplification can lead to biased supervision, poor calibration, and models that appear overconfident precisely when clinicians themselves hesitate. Recent work demonstrates that explicitly modeling rater-specific behavior and disagreement is essential to capture the true distribution of clinical opinions. Such approaches can improve robustness, interpretability, and deliver uncertainty estimates that align more closely with expert variability (Guan et al., 2018; Karimi et al., 2020; Zhang et al., 2023; Wong et al., 2022).

In Barrett's Esophagus (BE) surveillance, pathologists classify endoscopically derived esophageal biopsies into either nondysplastic BE (NDBE), low-grade dysplasia (LGD), or high-grade dysplasia (HGD). Dysplasia is the precursor stage of esophageal cancer and prompts intensified follow-up endoscopies when diagnosed. When confounders like inflammation are present, a pathologist may render the tissue biopsy to be indefinite for dysplasia, corresponding with an uncertain diagnosis. This diagnostic task of grading biopsies is known to be highly subjective and results in unwanted variation in patient treatment and outcome (Duits et al., 2015). International guidelines therefore advocate a second opinion. Large review efforts, such as the nationwide Dutch Esophageal Pathology Panel (DEPP), provide centralized expert assessment (van der Wel et al., 2019), routinely collecting multiple independent opinions per biopsy to mitigate this variability. These multi-rater labels offer a unique opportunity to study how deep learning models can capture expert disagreement instead of relying solely on a single consensus label.

In this paper, we leverage a multi-rater dataset collected by the DEPP, to study how the uncertainty could not only be modeled, but also used to train more accurate and better calibrated neural networks. In this dataset, twenty trained gastrointestinal pathologists from the DEPP independently reviewed more than 1500 BE biopsy cases. Each case consists of a hematoxylin and eosin (H&E) whole-slide image (WSI) and received multiple diagnostic opinions (median of eleven per case), demonstrating the extent of expert variability in real-world practice. A consensus reference diagnosis was established when at least 75% of participating pathologists agreed on the same grade. Cases lacking such agreement were discussed in dedicated consensus meetings until a final diagnosis was assigned. Figure 1 provides an example WSI and the corresponding participation matrix which pathologists contributed to each case. In addition, a separate set of 100 consecutive recently reviewed patient cases with full multi-rater annotations was reserved for final evaluation. Our contributions are as follows:

- we demonstrate that it is possible to train a "virtual rater" for each rater in the dataset, despite varying amounts of data;

- we have found that individual *intra*-rater predictions strongly correlate to consensus labels;

- we show—through different selection strategies—that it is possible to create a *virtual panel*, which is not only better at predicted grade than a network trained only on consensus label, but also that this virtual panel is better calibrated than the consensus one.

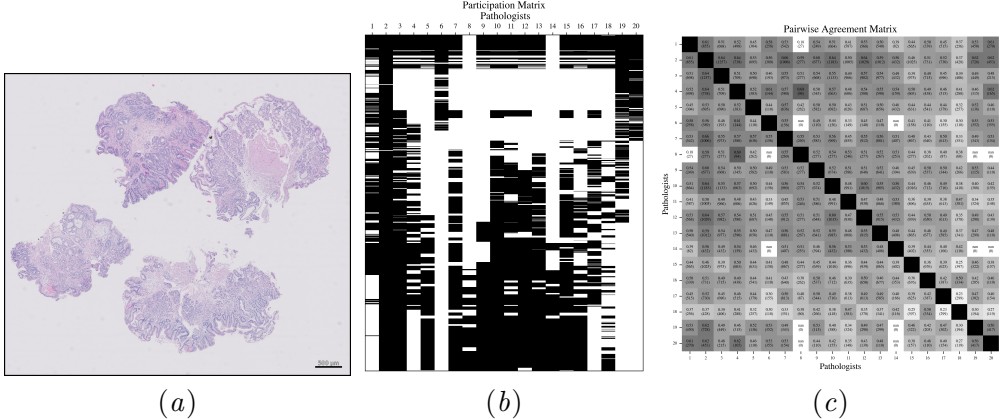

$(a)$             $(b)$             $(c)$

Figure 1: Overview of the dataset characteristics and rater variability. (a) Example WSI of a BE biopsy. (b) Visualization of pathologist participation per biopsy, illustrating the variable number of ratings per case. (c) Pairwise inter-rater agreement matrix, showing the agreement of diagnoses between each pair of expert pathologists using Krippendorff's alpha.

## 2. Related works

Deep learning methods that explicitly incorporate multiple expert labels have shown that diagnostic disagreement contains meaningful structure rather than mere noise. Classical probabilistic approaches such as Welinder et al. (2010) and Mnih and Hinton (2012) modeled annotators through confusion matrices and used EM-like procedures to jointly estimate latent true labels and annotator reliability, but these models cannot capture richer, image-dependent variability in rater behaviour. Guan et al. (2018) demonstrated that deep architectures which allocate a dedicated prediction head to each annotator and learn how to combine these heads can substantially outperform majority-vote training and EM-based label-aggregation baselines. Karimi et al. (2020) further highlighted that multi-rater disagreement should be viewed as structured label noise, motivating the use of soft-label supervision and noise-aware training strategies in medical imaging. More recently, Zhang et al. (2023) extended annotator modelling to dense prediction by jointly estimating a latent true segmentation and per-annotator confusion fields, allowing networks to capture spatially varying reliability and surpass both majority-vote and classical aggregation approaches. In neuropathology, Wong et al. (2022) trained "virtual neuropathologists" that emulate individual experts, showing that such models can match, and in some settings exceed, the diagnostic performance of the human raters whose behaviour they learn.

## 3. Methodology

### 3.1. Preliminary notation

In a typical supervised classification setting, we would have a dataset $\mathcal{D} \triangleq \{(x_n, y_n)\}_{n=1}^{N}$ with $x_n \in \mathbb{R}^{\Omega}$ denoting a single image ($\Omega$ some image space) and $y_n \in \mathbb{K}$ its label, with $\mathbb{K} \triangleq \{0 : \texttt{class-0}, \dots, K : \texttt{class-K}\}$ being the set of possible classes.

However, when multiple labels are available, such as in Figure 1, this typical notation falls short. In this paper, we refer to $\mathbb{A} \triangleq \{a_1, \ldots, a_A\}$ as the set of $A$ experts performing the ratings. For a specific sample $n$, we denote the individual rates (if any) as $^a y_n \in \widetilde{\mathbb{K}} \triangleq \mathbb{K} \cup \{-1\}$ ($-1$ denotes no rating for that sample, see Fig. 1.(b)), and $\bar{y}_n \in \mathbb{K}$ as the consensus rating (all samples have a consensus rating). We therefore have:

$$y_n \triangleq \{^a y_n\}_{a=1}^A \cup \{\bar{y}_n\}. \tag{1}$$

We can define a subset of samples rated by annotator $a$ as $\mathcal{D}_a \triangleq \{(x_n, y_n)|^a y_n \neq -1\} \subseteq \mathcal{D}$.

### 3.2. Training a virtual rater

From there, we can define the training of a neural network $\mathcal{N}(\cdot, \theta_a)$ based on that subset, to try to produce a "virtual rater", by minimizing:

$$\min_{\theta_a} \sum_{(x,y) \in \mathcal{D}_a} \mathcal{L}_{\text{CE}}(y, \mathcal{N}(x; \theta_a)) \tag{2}$$

where $\mathcal{N} : \mathbb{R}^\Omega \mapsto \triangle^K$ denotes some network architecture, $\theta_a$ its trainable parameters, and $\mathcal{L}_{\text{CE}}$ the standard cross-entropy loss.

This would produce a series of virtual raters $\theta_1, \ldots, \theta_A$, as well as $\bar{\theta}$ trained on the consensus labels $\bar{y}_n$. Performances could be evaluated in two manners: with respect to their own labels (*intra*-rater performances) and with respect to the consensus label (the *consensus* performances). This could be denoted as:

$$\begin{cases} \rho_a \propto \sum_{t \in \mathcal{T}} \rho(^a y_t, \mathcal{N}(x_t; \theta_a)) & \text{as } \textit{intra}\text{-rater performance,} \\ \bar{\rho}_a \propto \sum_{t \in \mathcal{T}} \rho(\bar{y}_t, \mathcal{N}(x_t; \theta_a)) & \text{as } \textit{consensus} \text{ performance,} \end{cases} \tag{3}$$

with $\rho(\cdot, \cdot)$ some metric and $\mathcal{T}$ some test set with labels for *every* rater.

### 3.3. From virtual raters to virtual panel

With now a set of $A$ virtual *raters* (assuming that the training converged), a virtual *panel* composed of several virtual raters could be defined. Several strategies could be used:

**Golden set**  $\mathring{\mathbb{A}} \subset \mathbb{A}$   A set of 5 "golden" raters , based on the best $\bar{\rho}_a$ performances could be used.

**Diverse set**  $\breve{\mathbb{A}} \subset \mathbb{A}$   Instead of picking the 5 "best" raters, and taking into account that some raters may be correlated between each others (either to undue influence, similar background/training, or other factors), one could aim to take not only pure performances into account, but also avoiding clusters of similar raters.

**Random set**  $\ddot{\mathbb{A}} \subset \mathbb{A}$ would simply consists to pick 5 virtual raters among the networks $\theta_a$ that converged.

Once a virtual panel is defined, it becomes trivial to combine the individual predictions to have not only a panel predictions, but an estimation of its uncertainty, that mimics the

dynamic of the real panel:

$$^a s \in \triangle^K \triangleq \mathcal{N}(x; \theta_a), \tag{4}$$

$$\overset{m}{p} \in \mathbb{K} \triangleq \arg\max_{k \in \mathbb{K}} \frac{1}{|\overset{m}{\mathbb{A}}|} \sum_{a \in \overset{m}{\mathbb{A}}} {}^a s, \tag{5}$$

$$\overset{m}{h} \in [0,1] \triangleq \frac{1}{|\overset{m}{\mathbb{A}}|} \mathbf{1}_{\left[\arg\max_k {}^a s = \overset{m}{p}\right]}, \tag{6}$$

with $\overset{m}{\cdot} \in \left\{ \overset{\circ}{\cdot}, \overset{\smile}{\cdot}, \overset{\cdot\cdot}{\cdot} \right\}$.

## 4. Experimental setup

### 4.1. Dataset details

The dataset that we used was collected over a period of 10 years (2015-2025) at the Amsterdam University Medical Centers coordinating the Dutch Esophageal Pathology Panel (DEPP), a national review panel of experts for the diagnosis of BE patients in the whole Netherlands, comprising 20 trained pathologists. In total, the panel reviewed 1577 BE biopsies originating from 1100 patients cases. Each pathologist reviewed a subset of cases, resulting in variable participation across raters. On average, each case received 11 independent diagnoses. Participation rates per rater ranged from 19.4% to 99.9%, with a mean of 50%. A visualization of biopsy-level participation is shown in Fig. 1(b).

Experts classify images into four dysplasia grades, following van der Wel (2019):

**NDBE** Non-dysplastic Barrett's Esophagus; low annual EAC risk (0.1–0.5%), monitored every 3–5 years.

**IND** Indefinite for dysplasia; unclear diagnosis due to confounders like inflammation.

**LGD** Low-grade dysplasia; moderate EAC risk (9–13%), treated endoscopically and monitored annually.

**HGD** High-grade dysplasia; pre-cancerous, requires endoscopic treatment and 3-month surveillance.

Since it remains unclear whether there is a recognizable and learnable pattern that leads pathologists to categorize cases as IND, the corresponding instances were excluded from the dataset and treated as missing values for the respective rater, decreasing the priorly mentioned participation rates. After filtering IND labels, 1432 cases remained for model development, with the class distribution shown in Table 1.

Consensus labels were computed using a 75% agreement threshold. Cases without such agreement were discussed in consensus meetings until a final diagnosis was assigned. A separate set of 100 consecutive recently reviewed patient cases (comprising 130 biopsies) with full multi-rater annotations was held out for final evaluation, with the class distribution shown in Table Table 1.

Each rater subset $\mathcal{D}_a$ is split into 5-folds for cross-validation: $\mathcal{D}_a = {}^a\mathcal{F}_1 \cup \cdots \cup {}^a\mathcal{F}_5$. As each set has a different size, and different distribution of dysplasia grades, we balanced as much as possible the sets based on "difficulty" of the cases. More details can be found in Appendix A.

Table 1: Distribution of dysplasia grades in the training set (1432 biopsies) and held-out evaluation set (130 biopsies).

| Dysplasia grade | Training set | Evaluation set |
|---|---|---|
| NDBE | 740 (51.7%) | 51 (39.2%) |
| LGD | 591 (41.3%) | 61 (46.9%) |
| HGD | 101 (7.1%) | 18 (13.8%) |
| **Total** | **1432** | **130** |

### 4.2. Metrics

The inter-rater reliability, displayed in Fig. 1(c), was computed as Krippendorff's alpha coefficient (Krippendorff, 1970; Krippendorff and Fleiss, 1978; Krippendorff, 2011), which inherently supports instances of sparse rating overlap.

Further performances of individual networks are assessed by the macro-averaged F1-score, which in our setting can be written as:

$$^a\text{F1} \equiv {}^a\text{F1}(^a\mathcal{T}) \triangleq \frac{1}{|\mathbb{K}|} \sum_{k \in \mathbb{K}} {}^a\text{F1}^{(k)}(^a\mathcal{T}), \tag{7}$$

where $^a\text{F1}_k(^a\mathcal{T})$ denotes the F1-score for class $k$ computed between the rater's original labels and the predictions of their virtual surrogate. Each class-specific F1-score is given by

$$\text{F1}^{(k)}(^a\mathcal{T}) \triangleq \frac{2 \times \text{Precision}^{(k)}(^a\mathcal{T}) \times \text{Recall}^{(k)}(^a\mathcal{T})}{\text{Precision}^{(k)}(^a\mathcal{T}) + \text{Recall}^{(k)}(^a\mathcal{T})} \tag{8}$$

To obtain final scores, we evaluated predictions across all five folds and reported average alpha and F1-scores per pathologist.

When training individual networks, convergence (or lack-thereof) for a window size of $T$ epochs is assessed using

$$\mathcal{A}_{val} \text{ converged} \iff \exists t_\ell \quad \frac{1}{T-1} \sum_{i=t_\ell}^{t_\ell+T} |\mathcal{A}_{\text{val}}[i+1] - \mathcal{A}_{\text{val}}[i]| < \varepsilon. \tag{9}$$

$\mathcal{L}_{\text{val}}$ and $\mathcal{A}_{\text{val}}$ denote the validation loss and validation accuracy at epoch $t$ respectively. Raters for which at least one of the five folds did not converge in terms of either validation accuracy or loss would be excluded from being in the final panel.

Finally assessing the calibration of aggregation methods, we calculate the expected calibration error (ECE) based on the accuracy and percentage agreement per bin $B_i$ with a bin size of 0.1:

$$\text{ECE} \triangleq \sum_{i=1}^{K_b} \frac{|B_i|}{N} |\text{acc}(B_i) - \text{conf}(B_i)|. \tag{10}$$

### 4.3. Implementation details

We adopt a weakly supervised framework based on attention-based multiple instance learning (ABMIL) (Ilse et al., 2018). WSIs were segmented and tiled into $224 \times 224$ patches at 1 mpp using the DLUP pipeline (Teuwen et al., 2024). Patch embedding were extracted with Virchow2 (Zimmermann et al., 2024), producing 2560-dimensional feature vectors and an $N \times 2560$ slide-level feature matrix.

The MIL network consists of a single-head attention module with a 16-dimensional hidden layer that aggregates patch features into a slide-level representation, followed by a linear prediction head for three-class grading (NDBE, LGD, HGD). Models were trained on these fixed features using categorical cross-entropy with a learning rate of $10^{-5}$ with a weight decay of $10^{-5}$. All experiments used 5-fold cross-validation, and we report performance averaged over folds.

We compare to an additional baseline (Guan et al., 2018), with the two variants Doctor-Net (DN) and Weighted DoctorNet (WDN). For consistency across experiments, the same model is used to extract patch embeddings, and the Inception Net v3 is replaced by the same ABMIL, to avoid overfitting while keeping variable elements as consistent as possible.

## 5. Results

### 5.1. Virtual raters

Table 3 summarizes both the intra and inter-rater agreement for each pathologists (which corresponds to Fig. 1(c)), the F1 score reached by the corresponding virtual rater, and if the training had converged or not. Finally, it displays the assigned cluster (for the diverse panel), and in which virtual panel they are assigned (if selected at all). reports the F1 score for the different models, evaluated both with respect to their own labels, but also with respect to the other panels (notably, consensus).

Figure 2 shows, for each rater, the relationship between their own F1-score (intra-rater assessment) and the F1-score with respect to the consensus. The color gradient displays the size of their own training set for each.

### 5.2. Virtual panels

In Table 3 we report the F1 score of the virtual panels, for different panel selections. The F1 score is computed across the different sets of annotations available, assessing generalization of the models for a different set of annotators. Finally, in the last column, we report the averaged F1 score across all annotation sets. In Figure 3 we can see the ECE for different panels selections strategies.

## 6. Discussion

**Virtual raters are trainable even with limited annotations.** One of the first main result, looking at Table 2 and Fig. 2 is that most networks did converge, even for those with less than 200 samples. It is also worth noting that less samples did not always result in lower performances, both in terms of intra-rater F1-score, or consensus F1-score. This is partly explainable to the consistency of each raters: WSI being notoriously difficult to

Table 2: Overall reliability scores per pathologist and their assigned cluster, including inter- and intra-rater reliability. Whether every fold of the virtual raters converged is indicated by an 'X'. Also indicated are the selected raters for the best-5 reliability strategy and the top-per-cluster strategy.

| | Rater Agreement | | | F1 | | | Panel selection | | |
|---|---|---|---|---|---|---|---|---|---|
| Pathologist | Inter | Intra | Overall | Intra | Converged | Cluster | Golden Å | Diverse Ä | Random Ä |
| 1 | 0.720 | 0.657 | 0.689 | 0.670 | √ | 1 | | | √ |
| 2 | 0.810 | 0.825 | 0.818 | 0.756 | √ | 1 | √ | √ | |
| 3 | 0.769 | 0.710 | 0.740 | 0.715 | √ | 1 | | | |
| 4 | 0.784 | 0.738 | 0.761 | 0.693 | √ | 1 | √ | | |
| 5 | 0.735 | 0.697 | 0.716 | 0.702 | √ | 2 | | | √ |
| 6 | 0.720 | 0.552 | 0.636 | 0.575 | √ | 2 | | | |
| 7 | 0.757 | 0.593 | 0.675 | 0.652 | √ | 2 | | | |
| 8 | 0.720 | 0.498 | 0.609 | 0.490 | | | | | |
| 9 | 0.731 | 0.696 | 0.713 | 0.674 | √ | 2 | | | √ |
| 10 | 0.732 | 0.774 | 0.753 | 0.774 | √ | 1 | √ | | |
| 11 | 0.710 | 0.671 | 0.690 | 0.663 | √ | 2 | | | |
| 12 | 0.753 | 0.723 | 0.738 | 0.752 | | | | | |
| 13 | 0.740 | 0.822 | 0.781 | 0.750 | √ | 2 | √ | √ | |
| 14 | 0.741 | 0.539 | 0.640 | 0.561 | √ | 4 | | √ | √ |
| 15 | 0.671 | 0.689 | 0.680 | 0.579 | √ | 0 | | √ | |
| 16 | 0.735 | 0.587 | 0.661 | 0.593 | √ | 3 | | √ | |
| 17 | 0.683 | 0.604 | 0.643 | 0.644 | √ | 2 | | | |
| 18 | 0.709 | 0.497 | 0.603 | 0.566 | | | | | |
| 19 | 0.775 | 0.708 | 0.741 | 0.692 | √ | 1 | | | |
| 20 | 0.778 | 0.716 | 0.747 | 0.643 | √ | 2 | √ | | √ |

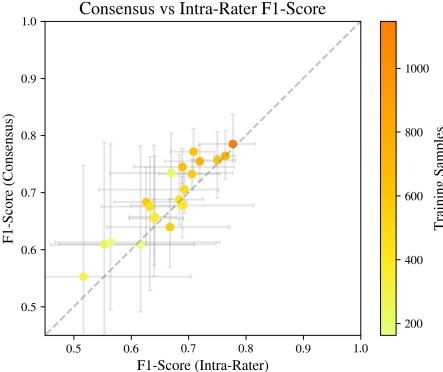

Figure 2: Comparison of virtual raters' F1-scores on consensus labels versus own labels. Each point represents mean performance with error bars indicating range of performances. Colors indicate the number of samples available.

grade, some raters may give different grades for the same slides if looking at it several times. This would translate into a decreased intra-rater F1-score.

**There is a very strong correlation between intra-rater F1-score and consensus F1-score.** Figure 2 clearly shows that the two are correlated, which is very promising for settings where a consensus label may not be available to assess the "reliability", "perfor-

Table 3: F1 score of the different virtual panels, with respect their own annotations, and the other annotations. $\overline{\mathrm{F1}}$ reports the average F1-score across all sets.

| Panel | F1 score (%) | | | | | |
|---|---|---|---|---|---|---|
| | $\overline{\mathcal{T}}$ | $\mathcal{T}_{\overset{\circ}{\mathbb{A}}}$ | $\mathcal{T}_{\check{\mathbb{A}}}$ | $\mathcal{T}_{\ddot{\mathbb{A}}}$ | $\mathcal{T}_{\mathbb{A}}$ | $\overline{\mathrm{F1}}$ |
| Consensus | 72.3±08.8 | 74.2±08.9 | 72.3±07.3 | 71.8±09.0 | 73.8±08.9 | 72.9 |
| Golden $\overset{\circ}{\mathbb{A}}$ | 75.8±07.9 | 76.3±08.5 | — | — | 78.3±08.0 | 76.8 |
| Diverse $\check{\mathbb{A}}$ | 77.5±08.0 | — | 75.8±08.6 | — | 77.3±08.5 | 76.9 |
| Random $\ddot{\mathbb{A}}$ | 75.0±08.4 | — | — | 76.7±08.1 | 77.4±08.4 | 76.4 |
| Full Panel $\mathbb{A}$ | 77.2±08.2 | 80.2±07.8 | 78.2±08.4 | 76.0±08.6 | 79.6±08.2 | 78.2 |
| DoctorNet | 74.5±08.5 | — | 77.5±08.5 | — | 78.1±08.4 | 76.7 |
| Weighted DoctorNet | 75.8±08.2 | — | 79.5±08.2 | — | 80.1±08.0 | 78.5 |

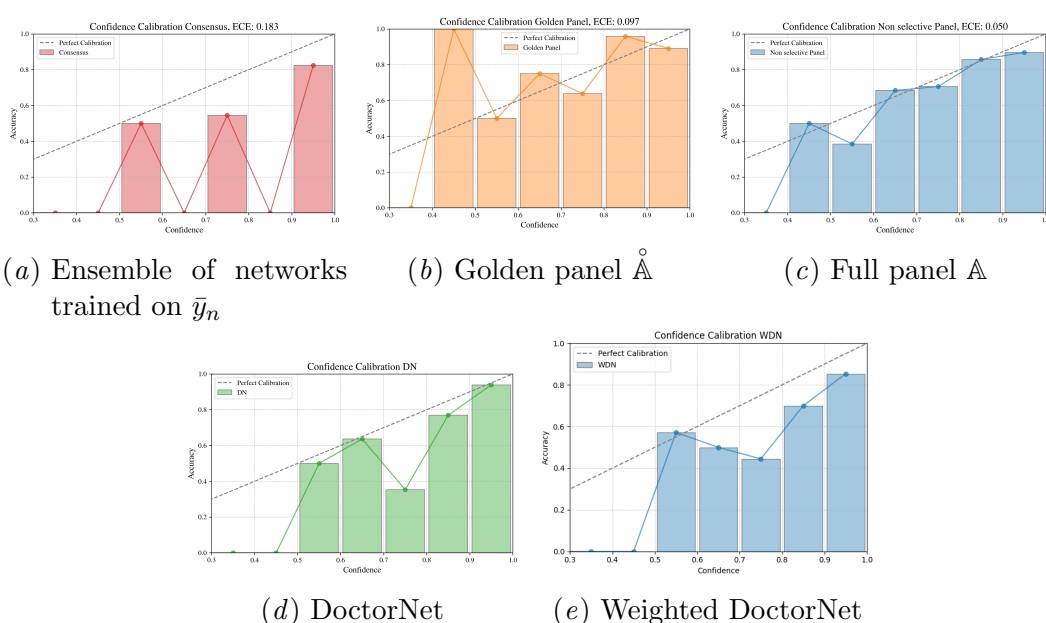

(a) Ensemble of networks trained on $\bar{y}_n$

(b) Golden panel $\overset{\circ}{\mathbb{A}}$

(c) Full panel $\mathbb{A}$

(d) DoctorNet

(e) Weighted DoctorNet

Figure 3: Expected class calibration error for different types of panels with percentage agreement as confidence estimates, showing correlation between confidence and actual performance as indicated by accuracy.

mance" of a single rater: it may be feasible, and informative enough, to use the internal consistency (intra-rater F1-score) as a proxy to make decisions.

**All virtual panel selection strategies perform better than networks trained solely on consensus annotations.** Even when comparing to the consensus annotations, those virtual panels perform better than the consensus based network.

**No selection/using the full panel for a virtual panel outperformed the more elaborate selection schemes.**  Perhaps one of the most surprising result of all, a virtual panel consistent of *all* networks that converged (regardless of their performances) outperformed the selection schemes that picked the 5 "best" networks.

**All virtual panels are better calibrated than the consensus based networks.** Likely one of the most clinically significant and useful result, the calibration (and hencerforth, the interpretability/reliability of the predictions) is better when using a virtual panel (this is true for all panel selections).

## 7. Conclusion

In this paper, we have shown that—even in settings with ambiguous ground-truth and lots of inter-rater variability—, multiple ratings can be leveraged in different ways. Individual networks can be trained for each rater, despite varying number of annotations available, and perhaps surprisingly initially, we have shown that there is a strong correlation between the intra-rater performances of the network, and the performances with respect to the consensus grading (the inter-rater part).

Moreover, we have shown that those individual networks can be combined into a virtual panel, that is not only more accurate than a network trained only on consensus data; but provides a better and more calibrated way to assess its confidence. This is of paramount importance in clinical settings, where confident but incorrect predictions could have devastating consequences for patients.

Perhaps most surprisingly, we have seen that using the full-panel of virtual raters—including the less accurate ones—led to the best performances. This is a very welcome results, as on top of the difficulty of getting more data in medical contexts, the question of inter-rater variability is always a difficult one to handle. This result seems to indicate—pending reproduction in similar settings—that simply using all of it without an attempt at curation may be a viable, if not better, strategy.

While future research is still required on how to best embed such a tool into a clinical practice, it is a promising avenue to either "boost" the expert panel on days where not many experts are available, or for countries and hospitals where such panels of experts are not available, especially in a timely manner (delayed diagnostic can lead to delayed treatment that could lead to much worse outcomes).

### Acknowledgments

This research was funded with a grant by the Maag Lever Darm Stichting (MLDS) with project number: MLDS WO 21-25.

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

## Appendix A. Dataset split for K-fold cross-validation

We assigned a discrete difficulty score to each instance, which was then used to balance the independently split datasets, extending the second approach. To define case difficulty, we used two metrics: the number of distinct diagnoses per case and the percentage agreement of rater diagnoses with the consensus. About half the cases showed unanimous ratings, while the rest had at least two distinct diagnoses (Table 1). Median percentage agreement is around 78 percent, with some cases exhibiting little or no agreement, making the mentioned two metrics useful assessors of case difficulty. Due to the absence of cases with only NDBE and HGD labels, the range of diagnoses was not considered.

We assessed percentage agreement by calculating the share of agreement between the rating of each available pathologist, formally:

$$\rho_n \triangleq \frac{1}{|A_n|} \sum_{a \in A_n} \mathbf{1}[{}^a y_n = \overline{y_n}], \tag{11}$$

where $\mathbf{1}$ is the indicator function. Additionally, the number of distinct diagnoses is formally denoted by:

$$\gamma_n \triangleq |\{k \mid \exists a \in A_n : {}^a y_n = k\}| \quad \text{with} \quad k \in \mathbb{K}. \tag{12}$$

To combine the two metrics into one difficulty score, we min-max-normalized $\gamma_n$ to be on the same scale as $\rho_n$ through:

$$\widetilde{\gamma}_n \triangleq \frac{\gamma_n - \min_i \gamma_i}{\max_i \gamma_i - \min_i \gamma_i}. \tag{13}$$

Scores were then combined by averaging the two individual measurements. Since the scales are inversed, we used the complement of agreement:

$$d_n \triangleq \frac{1}{2}\left(\widetilde{\gamma}_n + (1 - \rho_n)\right). \tag{14}$$

Finally, we discretized $d_n$ into three categories given the 33% and 66% quantiles, used to balance the dataset. We denote the categorical difficulty scores per case n as:

$$D_n \triangleq \begin{cases} \text{Low} & \text{if } \delta_n \leq Q(0.33) \\ \text{Medium} & \text{if } Q(0.33) < \delta_n \leq Q(0.66) \\ \text{High} & \text{if } \delta_n > Q(0.66). \end{cases} \tag{15}$$

