# OpenReview forum: "More is more: leveraging multi-rater information for whole slide images grading via virtual expert panel"
_MIDL.io/2026/Conference — MIDL 2026 Poster_

### Official Review · Reviewer_aXhm · 2026-01-06

**Confidence:** 4
**Preliminary Rating:** 4
**Final Rating:** 4

**Summary:**

This paper tackles whole-slide image (WSI) grading under substantial inter-observer variability. The authors train annotator-specific MIL models (“virtual raters”) and aggregate them into a “virtual expert panel.” Panel agreement (i.e., the fraction of models predicting the same label) is used as a confidence measure. The work compares multiple annotation/panel selection strategies (e.g., consensus-only, randomly sampled annotator subsets, and curated panels) in terms of classification performance and probability calibration.

**Strengths:**

1. Addresses a challenging and practically important setting. The work focuses on high inter-observer variability, a key bottleneck in pathology grading, and proposes a straightforward approach to leverage multi-rater information.

2. Careful experimental design across annotation strategies. Beyond a consensus-only baseline, the authors construct and evaluate several alternative annotation sets/panel selection strategies, enabling a more fine-grained analysis of how rater composition affects performance and calibration.

3. Well-defined problem with structured data. The dataset is clearly curated for a multi-rater evaluation protocol, which is well-aligned with the paper’s research question.

**Weaknesses:**

1. Practical deployability and computational cost: The approach requires training and running inference with
N annotator-specific models, which may be prohibitively expensive in real-world settings (latency, compute budget, maintenance).
2. Limited evidence for generalization. Experiments are conducted on a single dataset, making it difficult to assess robustness across institutions, grading schemes, and tasks.
3. Interpretability is unclear in the multi-model setting. ABMIL provides attention-based explanations, but it is not discussed how to present or consolidate interpretations across multiple virtual raters (e.g., disagreement regions, uncertainty-aware explanations).

**Detailed Comments:**

1. Figure readability. Figure 1(b,c) are too small to interpret. Consider removing the WSI thumbnail in (a) or redesigning the layout so that (b,c) are readable, with clearer axis labels and legends.

2. Missing textual explanations for notation/terms. Equations (4)–(6) are given, but key terms are not sufficiently explained in the main text.

3. Broader modeling baselines would strengthen the claim. To support generalization and fairness, it would be helpful to include additional MIL backbones beyond ABMIL (e.g., CLAM/TransMIL/DSMIL or other widely used WSI MIL variants), keeping the multi-rater strategy fixed.

**Justification Of Final Rating:**

The authors have effectively addressed the majority of the concerns raised during the review process. Specifically, the inclusion of additional baseline comparisons (e.g., DoctorNet and its variants) has significantly strengthened the validation of the proposed method. Regarding the request for an external validation dataset, I acknowledge the authors' justification that the uniqueness and complexity of the current dataset make immediate replication unfeasible. I accept this as a limitation common to specialized medical datasets rather than a fundamental flaw of the study.

The primary merit of this paper lies in its approach to the inherent subjectivity of histopathology. While inter-rater variability is a well-known challenge, computational studies that explicitly model this diversity remain scarce. This work provides compelling evidence for the efficacy of training individual models even when it involves including raters with relatively lower individual accuracy. By demonstrating that a diverse ensemble improves calibration and reliability, this study offers valuable insights that could shift the standard practice in learning from noisy medical annotations. Therefore, I maintain a positive rating for this work.

**Justification Of The Preliminary Rating:**

While the problem of observer variability in WSI is well-motivated, the paper has critical weaknesses as following: 1) High Overhead: Training $N$ separate models is computationally inefficient for real-world deployment. 2) Missing Baselines: Lack of comparison with standard multi-annotator methods makes it hard to distinguish the novelty from simple ensembling. 3) Limited Evaluation: The results are based on a single dataset, and the calibration analysis needs better clarification.

**Questions To Address In The Rebuttal:**

1. MIL method sensitivity: How sensitive are the results to the choice of ABMIL? Would the same gains hold with other MIL architectures?

2. Cross-dataset validation: Can the approach generalize to other known high-variability grading tasks (e.g., Gleason grading, Sydney gastritis classification)?

3. Clarify the intended takeaway of Figure 3: As presented, “ECE differs across panel selection strategies” alone is not necessarily meaningful. What actionable conclusion should the reader draw (e.g., Full panel consistently improves calibration; agreement is a reliable uncertainty proxy)?

4. Comparison to alternative ways of handling inter-observer variability: The paper would be significantly stronger with baselines such as:
[1] e.g., Who Said What: Modeling Individual Labelers Improves Classification (annotator-specific heads/embeddings).
[2] Soft-label / label-distribution learning baselines using the empirical label distribution across raters.

These comparisons are important to justify why training N separate models (and paying the inference cost) is preferable to a single shared model with annotator conditioning.

---

> ### Author Response · Authors · 2026-01-25
> **Response to Reviewer aXhm**
>
> > MIL method sensitivity: How sensitive are the results to the choice of ABMIL? Would the same gains hold with other MIL architectures?
>
> Due to time-constraints, this has not been studied thoroughly yet. We can only state that, empirically, we observed similar patterns, before sticking to ABMIL for consistency across experiments. A journal extension would definitely include a comparison across different encoders and backbones.
>
> > dataset validation: Can the approach generalize to other known high-variability grading tasks (e.g., Gleason grading, Sydney gastritis classification)?
>
> Plausibly, since the clinical challenges are similar. One of the main result, to us, is the ability to train individual networks, even when less annotations are available for some. The first step to test the generality of the findings would be to see if there is a similar correlation between intra-rater variability (embodied by the F1-score of its network) and inter-rater variability, as shown in Figure 2.
>
> > Clarify the intended takeaway of Figure 3: As presented, “ECE differs across panel selection strategies” alone is not necessarily meaningful. What actionable conclusion should the reader draw (e.g., Full panel consistently improves calibration; agreement is a reliable uncertainty proxy)?
>
> Indeed, the virtual panel agreement when used as a confidence score proved to be better calibrated, notably in the "in between" cases: where the probability is not close to 1, nor too low. Those likely represent some of the more difficult clinical cases and over-confidence (such as .8 predicted probability being correct only 40% of the time) could be quite detrimental.
>
> > Comparison to alternative ways of handling inter-observer variability: The paper would be significantly stronger with baselines such as: [1] e.g., Who Said What: Modeling Individual Labelers Improves Classification (annotator-specific heads/embeddings). [2] Soft-label / label-distribution learning baselines using the empirical label distribution across raters.
>
> See top-level response, where we compared to two additional baselines (DoctorNet and Weighted DoctorNet) from [1]. Additional baselines (and analysis of their trade-off, including interpretability and calibration) would be left for an eventual journal extension.
>
> > These comparisons are important to justify why training N separate models (and paying the inference cost) is preferable to a single shared model with annotator conditioning.
>
> See top-level response.

---

### Official Review · Reviewer_tL58 · 2026-01-13

**Confidence:** 4
**Preliminary Rating:** 3
**Final Rating:** 3

**Summary:**

This work proposes “virtual raters” for WSI grading. A MIL classifier is trained per pathologist using that rater's labels, and these models are then combined into a virtual expert panel by aggregating their predictive distributions and using panel agreement as confidence. Experiments on a Barretts Esophagus multi-rater dataset show that panel aggregation outperforms training on consensus labels alone and improves calibration.

**Strengths:**

- The paper shows a simple and practical idea with clear clinical motivation. They are treating disagreement as signal and turning multiple experts into an ensemble-like panel. This is intuitive and easy to adopt in real multi-rater settings.
- Evaluating ECE is clinically relevant, because overconfident wrong predictions are a real risk. This strengthens the narrative beyond accuracy.
- Method and paper structure are mostly coherent, definitions and the experimental pipeline (ABMIL + fixed Virchow2 features, 5-fold CV) are described sufficiently to reproduce at high level and is well in line with the state-of-the-art

Value to community: the paper makes a compelling case that using more raters can improve both performance and reliability for difficult clinical cases. To my knowledge, this is an understudied topic in computational pathology.

**Weaknesses:**

- The main weakness of the paper is that the baseline coverage is incomplete for a multi-rater learning paper. The comparisons are largely internal (e.g. consensus only vs panel variants). It would be much stronger to compare against established multi-annotator aggregation or learning strategies (e.g. Guan et al "Who Said What: Modeling Individual Labelers Improves Classification", AAAI 2018. Another baseline could be e.g., Whitehill et al "Whose Vote Should Count More: Optimal Integration of Labels from Labelers of Unkown Experts", NIPS 2009).
- Method section readability "is an index fight". The notation is heavy and may obscure the core idea. A simpler description would improve accessibility.
- Limited experiments. The evaluation appears to use a single dataset from the authors institution and a held-out set drawn from the same source. There is neither a second dataset used for experiments used nor an external validation conducted for the same task. The paper risks being seen that observed gains could be dataset / institutional specific and generalization is questionable.

**Detailed Comments:**

- there is a space missing in caption of Table 3 between "F1" and "reports": "the other annotations. F1reports the average F1-score across all sets."
- did the authors use macro F1 metric instead of standard metric F1? Note that the label distributions are very uneven (especially for class HGD, as in Tab. 1). Common F1 is vulnerable to uneven label distributions and the delta gains of the proposed outcomes might risk being smaller in such cases. Moreover, can there be AUROC values provided as well, since reporting F1 scores only is threshold-dependent in classification.
- Lots of text parts seem to be LLM generated, which sometimes makes the paper read like a chat response rather than a scientific work (e.g. lots of "--" and brackets (...), etc.)
- I dont understand what the symbol with the triangle over the equation symbol is, e.g. Eq. 7, can the authors clarify this?
- In Eq. 3, the authors use the proportionality sign, while there is a metric on the left hand side. Can the authors clarify this Eq. and why they used a proportionality symbol there (i rather know that from probabilities).

**Justification Of Final Rating:**

The authors have addressed all questions regarding methodology notation. While I think the multi-rater topic is interesting for the computational pathology community, the proposed method has been evaluated on only one dataset and with one comparable method and thus lacks generalization and risks overstating its practical usefulness beyond this specific experimental setting. Thus, I keep my preliminary score.

**Justification Of The Preliminary Rating:**

The paper presents a timely and clinically very relevant topic. However, in its current form, it can be only seen as: we have this dataset and try different obvious strategies to verify "what label usage work best" and there is not too much more insights. Unfortunately, it is unknown if the dataset will also be available to the community, which would a big value to the community. The paper can significantly benefit from more baselines to compare with, another dataset or external validation and simplifying methodology section.

**Questions To Address In The Rebuttal:**

For example:
- include more comparable methods, see "Weaknesses"
- simplify the "index-fight" in the methodology section
- increase experiments for external validation or another multi-rater dataset.
- provide exact description of which F1 metric is used, and append AUROC metric results to the Appendix.

---

> ### Author Response · Authors · 2026-01-25
> **Response to Reviewer tL58**
>
> > Lots of text parts seem to be LLM generated, which sometimes makes the paper read like a chat response rather than a scientific work (e.g. lots of "--" and brackets (...), etc.)
>
> No LLM was used in any form for this paper. The last author has simply been a pedant for years—often facilitated by [1]—and pre-dating the current LLM-craze. But we acknowledge that the writing of that manuscript had not been as smooth as it could have been, and work to polish it (taking reviewers feedback) it underway.
>
> > I dont understand what the symbol with the triangle over the equation symbol is, e.g. Eq. 7, can the authors clarify this?
>
>  $\triangleq$ is used here to mean a _definition_, which we try to distinguish from results from calculation. Other convetions may use $:=$ or $\stackrel{\text{def}}{=}$. The $\triangleq$ was preferred over the later due to its frequent use on paper/whiteboard.
>
> > In Eq. 3, the authors use the proportionality sign, while there is a metric on the left hand side. Can the authors clarify this Eq. and why they used a proportionality symbol there (i rather know that from probabilities).
>
> The proportionality sign was used there to avoid writing any possible normalization factor.
>
> > include more comparable methods, see "Weaknesses"
>
> See top-level response, where we included
>
> > simplify the "index-fight" in the methodology section
>
> We are working to streamline the notation. Note that a lot of the notation appeared during our discussions, where the need for a very precise notation felt to understand each-others clearly and unambiguously. Unfortunately the uniqueness of the dataset and challenge of the task complexity the notation at many different levels.
>
> > provide exact description of which F1 metric is used, and append AUROC metric results to the Appendix.
>
> Macro-averaged F1 is used. Macro-averaging ensures that all classes contribute equally to the metric, regardless of their frequency, mitigating the (varying) class imbalance present in the datasets. To obtain final scores, we evaluated predictions across all five folds.

---

### Official Review · Reviewer_Fa2c · 2026-01-17

**Confidence:** 4
**Preliminary Rating:** 3
**Final Rating:** 4

**Summary:**

This paper addresses the issues of poor calibration and overconfident predictions that comes with a consensus label, i.e. compressing many expert diagnosis/gradings into a single label. The proposed multi-rater learning method models individual rater behavior. These virtual raters provide individual predictions which are then aggregated to provide the final prediction. The variation among the virtual raters also improves the quality of uncertainty estimates and provides better calibrated predictions.

**Strengths:**

1. The proposed virtual panel outperforms a standard model trained on the consensus label (which tends to overfit to the harder labels of the consensus). The virtual panel appears statistically more robust.

2. The virtual panel provides a much better calibration compared to the consensus model. Example: when there is high variance among the virtual raters, the model's confidence drops accordingly rather than being overconfident (as seen in the consensus label method).

3. Adding more raters (including those perceived as noisy) to the ensemble tends to improve performance.

4. Instead of incorporating complex loss functions, the proposed method achieves calibration naturally. The variance of the virtual panel acts as a proxy to the aleatoric uncertainty of the image.

5. The method is validated on the highly challenging DEPP dataset where each image has 11 expert panels thereby showing high inter-observer variability.

**Weaknesses:**

1. The proposed method involves training N separate models. Assuming there is no saturation point with respect to the number of annotators, N - wouldn't this be computationally expensive ? How would this method compare against a multi-head architecture having one backbone and N heads ?

2. The evaluation metric is accuracy against the Consensus Label (ground-truth) which itself is flawed and tends to be overconfident. Did the authors explore other evaluation metrics such as longitudinal outcome prediction, etc . ?

3. For the DEPP dataset: inference would involve running the full ensemble 11 times per slide. In terms of clinical use, this would increase latency and adversely affect real-time clinical workflows.

4. Given that the DEPP dataset is extremely small, how do the authors prevent overfitting while training the model ? This is unclear from the paper.

5. The computational cost of training is not discussed - this would be important from an implementation standpoint.

**Detailed Comments:**

no further comments to add

**Justification Of Final Rating:**

The authors have satisfactorily addressed the review comments and made changes to the paper that make it suitable for publication to the MIDL conference. After careful deliberation, I have decided to increase my original rating to 4.

**Justification Of The Preliminary Rating:**

The method proposed in this paper is promising and addresses the issue of having multiple raters for assessing a limited dataset and providing a calibrated expert rating. There are some unanswered questions that have come up while reviewing the paper which prevented this paper from getting a higher score. I look forward to the authors' comments.

**Questions To Address In The Rebuttal:**

please refer to the comments in the 'weaknesses' section and address them.

---

> ### Author Response · Authors · 2026-01-25
> **Response to Reviewer Fa2c**
>
> > The proposed method involves training N separate models. Assuming there is no saturation point with respect to the number of annotators, N - wouldn't this be computationally expensive ? How would this method compare against a multi-head architecture having one backbone and N heads ?
>
> Please see to our top-level response. To complete it, yes, the computational complexity of the head is O(N) compared to a multi-head network, but it remains marginal compared to any other time-consuming step (be it computational, or the tedious multi-days slicing/staining/scanning of the biopsy).
>
> > The evaluation metric is accuracy against the Consensus Label (ground-truth) which itself is flawed and tends to be overconfident. Did the authors explore other evaluation metrics such as longitudinal outcome prediction, etc . ?
>
> Evaluation was conducted against several labels, including a full panel majority vote and labels derived from selected five-pathologist-panels (see Table 3). Given the abscence of longitudinal data, we thereby try to reduce label noise by including different reference standards. We are still looking at better ways to convey this in the paper (this is also what creates the complexity in the notation mentioned by Reviewer tL58).
>
> > For the DEPP dataset: inference would involve running the full ensemble 11 times per slide. In terms of clinical use, this would increase latency and adversely affect real-time clinical workflows.
>
> > The computational cost of training is not discussed - this would be important from an implementation standpoint.
>
> We addressed this in the top-level response, and we will improve the manuscript in a camera-ready version ; making good use of the additional two pages allowed there. Moreover, the documentation of the public code will be improved, to clarify the computational requirements and expected runtimes.

---

### Author Response · Authors · 2026-01-25
**Top-level response**

We thank all reviewers for their kind and warm reviews.

While we will respond to each reviewer individually, this top-level response will be used to address some shared questions and request for clarification.

---

Notably, and we apologize for this, we had completely skipped in the submitted paper any discussion on computational cost. For this task, we had used a two-stages pipeline, with Virchow2 extracting patch-level features once (the patch encoding are reused across all methods). This process takes about 3 hours, without any parallelization (about 5-20 seconds per slide, which varies depending on the number on non-empty patches detected), for our dataset (1150 slides). The per-rater networks are then trained on the encodings, and we had sticked to ABMIL for all experiments (this was motivated by parallel research done within the group, were it was found empirically to be a simple and effective method). _Those individual rater networks take about 20 minutes to train, and inference takes around 50ms per network._

In short, in all experiments and design done, the additional cost between training one network and 10 is marginal (20 networks is basically overnight, which is a typical flow for many researchers), particularly at inference. From our perspective, this means that we can get better confidence estimation "for free", practically speaking.

---

We also wish to stress that we would be absolutely thrilled to have a second dataset to assess the generability of our findings. However, the value and uniqueness of the dataset (and in our view of the findings themselves), also make it difficult to collect a second dataset with similar qualities:

The dataset is composed of 1150 consecutive BE biopsy revision requests from 39 hospitals, that were assessed in the Dutch Esophageal Pathology Panel between February 2015 and December 2024 in The Netherlands. The panel was initially established in 2015 with five core members and has undergone constant changes in composition over time with pathologists both joining and leaving, and in total, 20 different pathologists have participated. As a requirement to become a panel member, all pathologists completed multiple stringent training programs, designed with benchmark criteria, to ensure reliable and consistent histopathological interpretation, with interactive feedback sessions being part of this.

Each BE biopsy revision case is independently assessed by panel pathologists using a digital slide viewer, remaining blinded to both the referring hospital pathology sign-out report and the assessment of other panel pathologists. The panel diagnosis was defined as an interpathologist agreement above 75% among the participating panel pathologists. The minimum number of pathologists required to achieve >75% agreement was four. If the overall agreement between the pathologists fell below this threshold, cases were discussed in a monthly online consensus meeting to reach final consensus on the diagnosis. On average, each case received 11 independent diagnoses. Participation rates per rater ranged from 19.4% to 99.9%, with a mean of 50%.

As a result of this process, this panel and dataset is unique in the world, as expertise is quantified and subject to both evolving diagnostic insights through time and evolving composition of the data interpreters. This pathologists panel is simultaneously being homogeneous due to training and monthly feedback session and heterogeneous due to variation in interpretation, background of specialty training and exposure to daily workload and environment in different parts of the Netherlands. A similar data set does not exist and **acquiring any data set alike would not be feasible within a time frame of many years.**

While we would ultimately wish to make the dataset available to other researchers, there is currently no timeline where this would become a reality. Following the reviews we had an internal discussion pondering if releasing the patch encodings and anonymized ratings would be feasible, but this would require to be discussed and approved by the privacy and ethical committee of the hospital first.

Future works and extensions will find the most relevant and similar datasets and evaluate thoroughly on them, but doing this properly will take time and we already believed that the current insights are worth sharing with the community.

---

> ### Author Response · Authors · 2026-01-25
> **Top-level comment (cont.)**
>
> Following the suggestion of several reviewers, we did run comparison on additional baselines [1]. Their implementation relies on the same feature extraction (Virchow2). DoctorNet initially was implemented with an Inception Net v3, but we decided to reuse the same ABMIL as other methods, both to avoid overfitting on our smaller dataset, while maintaining more comparable results. Ultimately, the key methodological development from DoctorNet and Weighted Doctor Net is how they handle the multiple labels.
>
> Both methods proved to perform well in pure classification performance:
>
> | Panel | $\overline{\mathcal T}$ | $\mathcal T_{\circ \mathbb A}$ | $\mathcal T_{\breve{\mathbb A}}$ | $\mathcal T_{\ddot{\mathbb A}}$ | $\mathcal T_{\mathbb A}$ | $\overline{\text{F1}}$ |
> | :--- | :---: | :---: | :---: | :---: | :---: | :---: |
> | Consensus | 72.3 ± 8.8 | 74.2 ± 8.9 | 72.3 ± 7.3 | 71.8 ± 9.0 | 73.8 ± 8.9 | 72.9 |
> | Golden $\circ \mathbb A$ | 75.8 ± 7.9 | 76.3 ± 8.5 | NA | NA | 78.3 ± 8.0 | 76.8 |
> | Diverse $\breve{\mathbb A}$ | 77.5 ± 8.0 | NA | 75.8 ± 8.6 | NA | 77.3 ± 8.5 | 76.9 |
> | Random $\ddot{\mathbb A}$ | 75.0 ± 8.4 | NA | NA | 76.7 ± 8.1 | 77.4 ± 8.4 | 76.4 |
> | Full Panel $\mathbb A$ | 77.2 ± 8.2 | 80.2 ± 7.8 | 78.2 ± 8.4 | 76.0 ± 8.6 | 79.6 ± 8.2 | 78.2 |
> | DoctorNet | 74.5 ± 8.5 | NA | 77.5 ± 8.5 | NA | 78.1 ± 8.4 | 76.7 |
> | WeightedDoctorNet | 75.8 ± 8.2 | NA | 79.5 ± 8.2 | NA | 80.1 ± 8.0 | 78.5 |
>
> However, when looking at the calibration of the methods (added Fig. 3 in the Paper), we can see it is not as good as the virtual panels (notably the accuracy of the 0.7-0.8 bin is significantly lower, around 0.4 accuracy). Considering that this likely impacts most of the more difficult, less clear-cut clinical cases, it is a notable draw-back for those two methods.
>
> Moreover, when inspecting the final model of Weighted DoctorNet (that fine-tunes a unique weight for each rater), we found that WDN did ultimately rely only on a few annotators, by giving them an outsized weight, and a very low for the other raters [2]. This highlights how the method relies directly on the quality and consistency of the consensus label. On the contrary, the multi-network/per-rater rater we studied did not explicitly optimize with respect to that consensus label.
>
> We find this key difference in the working of the methods very interesting, and a valuable insight to keep in mind or guide future research around this topic.
>
>
> [1] Guan, M., Gulshan, V., Dai, A., & Hinton, G. (2018, April). Who said what: Modeling individual labelers improves classification. In Proceedings of the AAAI conference on artificial intelligence (Vol. 32, No. 1).
> [2] `0.0009503824 0.7140262 0.005313257 0.008792963 0.0005944913 0.20234218 0.0011595053 0.00068232365 0.000575638 0.050800037 0.00083758135 0.0020768389 0.0023317572 0.00075822975 0.0050401306 0.0005433953 0.00056192646 0.0005818316 0.000850273 0.0011811104 `

---

### Author Rebuttal · Authors · 2026-01-25

**Rebuttal:**

See updted manuscript and direct comments

**Supporting Material:**

/attachment/9c86fbc042485f3a8bdeedfb88686b6326cd0d81.pdf

---

### Meta-Review · Area_Chair_SGcq · 2026-02-08

**Recommendation:** Accept (Poster)
**Confidence:** 4

**Metareview:**

The authors have successfully addressed the comments of the reviewers, and in general, the study has a unique element of a dataset that most likely does not exist anywhere else in the world. I therefore recommend the acceptance of this paper.

---

### Decision · Program_Chairs · 2026-02-14

Accept (Poster)